# GENERATIVE TEACHING NETWORKS: ACCELERATING NEURAL ARCHITECTURE SEARCH BY LEARNING TO GENERATE SYNTHETIC TRAINING DATA

## ABSTRACT

This paper investigates the intriguing question of whether we can create learning algorithms that automatically generate training data, learning environments, and curricula in order to help AI agents rapidly learn. We show that such algorithms are possible via Generative Teaching Networks (GTNs), a general approach that is, in theory, applicable to supervised, unsupervised, and reinforcement learning, although our experiments only focus on the supervised case. GTNs are deep neural networks that generate data and/or training environments that a learner (e.g. a freshly initialized neural network) trains on for a few SGD steps before being tested on a target task. We then differentiate *through the entire learning process* via meta-gradients to update the GTN parameters to improve performance on the target task. GTNs have the beneficial property that they can theoretically generate any type of data or training environment, making their potential impact large. This paper introduces GTNs, discusses their potential, and showcases that they can substantially accelerate learning. We also demonstrate a practical and exciting application of GTNs: accelerating the evaluation of candidate architectures for neural architecture search (NAS), which is rate-limited by such evaluations, enabling massive speed-ups in NAS. GTN-NAS improves the NAS state of the art, finding higher performing architectures when controlling for the search proposal mechanism. GTN-NAS also is competitive with the overall state of the art approaches, which achieve top performance while using orders of magnitude less computation than typical NAS methods. Speculating forward, GTNs may represent a first step toward the ambitious goal of algorithms that generate their own training data and, in doing so, open a variety of interesting new research questions and directions.

## 1 INTRODUCTION AND RELATED WORK

Access to vast training data is now common in machine learning. However, to effectively train neural networks (NNs) does not require using *all available* data. For example, recent work in curriculum learning (Graves et al., 2017), active learning (Konyushkova et al., 2017; Settles, 2010) and core-set selection (Sener & Savarese, 2018; Tsang et al., 2005) demonstrates that a surrogate dataset can be created by intelligently sampling a subset of training data, and that such surrogates enable competitive test performance with less training effort. Being able to more rapidly determine the performance of an architecture in this way could particularly benefit architecture search, where training thousands or millions of candidate NN architectures on full datasets can become prohibitively expensive. From this lens, related work in learning-to-teach has shown promise. For example, the learning to teach (L2T) (Fan et al., 2018) method accelerates learning for a NN learner (hereafter, just *learner*) through reinforcement learning, by learning how to subsample mini-batches of data.

A key insight in this paper is that the surrogate data need not be drawn from the original data distribution (i.e. they may not need to resemble the original data). For example, humans can learn new skills from reading a book or can prepare for a team game like soccer by practicing skills, such as passing, dribbling, juggling, and shooting. This paper investigates the question of whether we can train a data-generating network that can produce *synthetic* data that effectively and efficiently teaches a target task to a learner. Related to the idea of generating data, Generative Adversarial Networks (GANs) can produce impressive high-resolution images (Goodfellow et al., 2014; Brock

et al., 2018), but they are incentivized to mimic real data (Goodfellow et al., 2014), instead of being optimized to teach learners *more* efficiently than real data.

Another approach for creating surrogate training data is to treat the training data itself as a hyper-parameter of the training process and learn it directly. Such learning can be done through meta-gradients (also called hyper-gradients), i.e. differentiating through the training process to optimize a meta-objective. This approach was described in Maclaurin et al. (2015), where 10 synthetic training images were learned using meta-gradients such that when a network is trained on these images, the network's performance on the MNIST validation dataset is maximized. In recent work concurrent with our own, Wang et al. (2019b) scaled this idea to learn 100 synthetic training examples. While the 100 synthetic examples were more effective for training than 100 original (real) MNIST training examples, we show that it is difficult to scale this approach much further without the regularity across samples provided by a generative architecture (Figure 2b, green line).

Being able to very quickly train learners is particularly valuable for neural architecture search (NAS), which is exciting for its potential to automatically discover high-performing architectures, which otherwise must be undertaken through time-consuming manual experimentation for new domains. Many advances in NAS involve accelerating the evaluation of candidate architectures by training a predictor of how well a trained learner would perform, by extrapolating from previously trained architectures (Luo et al., 2018; Liu et al., 2018a; Baker et al., 2017). This approach is still expensive because it requires many architectures to be trained and evaluated to train the predictor. Other approaches accelerate training by sharing training across architectures, either through shared weights (e.g. as in ENAS; Pham et al. (2018)), or Graph HyperNetworks (Zhang et al., 2018).

We propose a scalable, novel, meta-learning approach for creating synthetic data called Generative Teaching Networks (GTNs). GTN training has two nested training loops: an inner loop to train a learner network, and an outer-loop to train a generator network that produces synthetic training data for the learner network. Experiments presented in Section 3 demonstrate that the GTN approach produces synthetic data that enables much faster learning, speeding up the training of a NN by a factor of 9. Importantly, the synthetic data in GTNs is not only agnostic to the weight initialization of the learner network (as in Wang et al. (2019b)), but is also agnostic to the learner's *architecture*. As a result, GTNs are a viable method for *accelerating evaluation* of candidate architectures in NAS. Indeed, controlling for the search algorithm (i.e. using GTN-produced synthetic data as a drop-in replacement for real data when evaluating a candidate architecture's performance), GTN-NAS improves the NAS state of the art by finding higher-performing architectures than comparable methods like weight sharing (Pham et al., 2018) and Graph HyperNetworks (Zhang et al., 2018); it also is competitive with methods using more sophisticated search algorithms and orders of magnitude more computation. It could also be combined with those methods to provide further gains.

One promising aspect of GTNs is that they make very few assumptions about the learner. In contrast, NAS techniques based on shared training are viable only if the parameterizations of the learners are similar. For example, it is unclear how weight-sharing or HyperNetworks could be applied to architectural search spaces wherein layers could be either convolutional or fully-connected, as there is no obvious way for weights learned for one layer type to inform those of the other. In contrast, GTNs are able to create training data that can generalize between such diverse types of architectures.

GTNs also open up interesting new research questions and applications to be explored by future work. Because they can rapidly train new architectures, GTNs could be used to create NNs *on-demand* that meet specific design constraints (e.g. a given balance of performance, speed, and energy usage) and/or have a specific subset of skills (e.g. perhaps one needs to rapidly create a compact network capable of three particular skills). Because GTNs can generate virtually any learning environment, they also one day could be a key to creating AI-generating algorithms, which seek to bootstrap themselves from simple initial conditions to powerful forms of AI by creating an open-ended stream of challenges (learning opportunities) while learning to solve them (Clune, 2019).

## 2 METHODS

The main idea in GTNs is to train a data-generating network such that a learner network trained on data it *rapidly* produces high accuracy in a target task. Unlike a GAN, here the two networks cooperate (rather than compete) because their interests are aligned towards having the learner perform well on the target task when trained on data produced by the GTN. The generator and the

learner networks are trained with meta-learning via nested optimization that consists of inner and outer training loops (Figure 1a). In the inner-loop, the generator $G(z, y)$ takes Gaussian noise ($z$) and a label ($y$) as input and outputs synthetic data ($x$). Optionally, the generator could take only noise as input and produce both data and labels as output (Appendix F). The learner is then trained on this synthetic data for a fixed number of inner-loop training steps with any optimizer, such as SGD or Adam (Kingma & Ba, 2014): we use SGD with momentum in this paper. SI Equation 1 defines the inner-loop SGD with momentum update for the learner parameters $\theta_t$. We sample $\mathbf{z}_t$ (noise vectors input to the generator) from a unit-variance Gaussian and $\mathbf{y}_t$ labels for each generated sample) uniformly from all available class labels. Note that both $\mathbf{z}_t$ and $\mathbf{y}_t$ are batches of samples. We can also learn a curriculum directly by additionally optimizing $\mathbf{z}_t$ directly (instead of sampling it randomly) and keeping $\mathbf{y}_t$ fixed throughout all of training.

The inner-loop loss function $\ell_{\text{inner}}$ can be cross-entropy for classification problems or mean squared error for regression problems. Note that the inner-loop objective does not depend on the outer-loop objective and could even be parameterized and learned through meta-gradients with the rest of the system (Houthooft et al., 2018). In the outer-loop, the learner $\theta_T$ (i.e. the learner parameters trained on *synthetic* data after the $T$ inner-loop steps) is evaluated on the real *training* data, which is used to compute the outer-loop loss (aka meta-training loss). The gradient of the meta-training loss with respect to the generator is computed by backpropagating through the entire inner-loop learning process. While computing the gradients for the generator we also compute the gradients of hyper-parameters of the inner-loop SGD update rule (its learning rate and momentum), which are updated after each outer-loop at no additional cost. To reduce memory requirements, we leverage gradient-checkpointing (Griewank & Walther, 2000) when computing meta-gradients. The computation and memory complexity of our approach can be found in Appendix D.

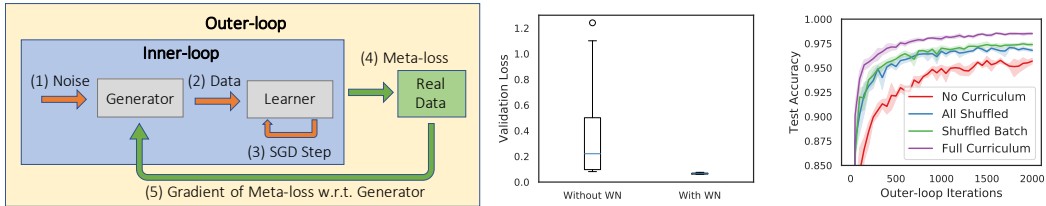

(a) Overview of Generative Teaching Networks    (b) GTN stability with WN    (c) GTN curricula comparison

Figure 1: (a) Generative Teaching Network (GTN) Method. The numbers in the figure reflect the order in which a GTN is executed. Noise is fed as an input to the Generator (1), which uses it to generate new data (2). The learner is trained (e.g. using SGD or Adam) to perform well on the generated data (3). The trained learner is then evaluated on the real training data in the outer-loop to compute the outer-loop meta-loss (4). The gradients of the generator parameters are computed w.r.t. to the meta-loss to update the generator (5). Both a learned curriculum and weight normalization substantially improve GTN performance. (b) Weight normalization improves meta-gradient training of GTNs, and makes the method much more robust to different hyperparameter settings. Each boxplot reports the final loss of 20 runs obtained *during* hyperparameter optimization with Bayesian Optimization (lower is better). (c) shows a comparison between GTNs with different types of curricula. The GTN method with the most control over how samples are presented performs the best.

A key motivation for this work is to generate synthetic data that is learner agnostic, i.e. that generalizes across different potential learner architectures and initializations. To achieve this objective, at the beginning of each new outer-loop training, we choose a new learner architecture according to a predefined set and randomly initialize it (details in Appendix A).

**Meta-learning with Weight Normalization.** Optimization through meta-gradients is often unstable (Maclaurin et al., 2015). We observed that this instability greatly complicates training because of its hyperparameter sensitivity, and training quickly diverges if they are not well-set. Combining the gradients from Evolution Strategies (Salimans et al., 2017) and backpropagation using inverse variance weighting (Fleiss, 1993; Metz et al., 2019) improved stability in our experiments, but optimization still consistently diverged whenever we increased the number of inner-loop optimization steps. To mitigate this issue, we introduce applying weight normalization (Salimans & Kingma, 2016) to stabilize meta-gradient training by normalizing the generator and learner weights. Instead

of updating the weights ($W$) directly, we parameterize them as $W = g \cdot V/\|V\|$ and instead update the scalar $g$ and vector $V$. Weight normalization eliminates the need for (and cost of) calculating ES gradients and combining them with backprop gradients, simplifying and speeding up the algorithm.

We hypothesize that weight normalization will help stabilize meta-gradient training more broadly, although future work is required to test this hypothesis in meta-learning contexts besides GTNs. The idea is that applying weight normalization to meta-learning techniques is analogous to batch normalization for deep networks (Ioffe & Szegedy, 2015). Batch normalization normalizes the forward propagation of activations in a long sequence of parameterized operations (a deep NN). In meta-gradient training both the activations and weights result from a long sequence of parameterized operations and thus both should be normalized. Results in section 3.1 support this hypothesis.

**Learning a Curriculum with Generative Teaching Networks.** Previous work has shown that a learned curriculum can be more effective than training from uniformly sampled data (Graves et al., 2017). A curriculum is usually encoded with indexes to samples from a given dataset, rendering it non-differentiable and thereby complicating the curriculum's optimization. With GTNs however, a curriculum can be encoded as a series of input vectors to the generator (i.e. instead of sampling the $\mathbf{z}_t$ inputs to the generator from a Gaussian distribution, a sequence of $\mathbf{z}_t$ inputs can be learned). A curriculum can thus be learned by differentiating through the generator to optimize this sequence (in addition to the generator's parameters). Experiments confirm that GTNs more effectively teach learners when optimizing such a curriculum (Section 3.2).

**Accelerating NAS with Generative Teaching Networks.** Since GTNs can accelerate learner training, we propose harnessing GTNs to accelerate NAS. Rather than evaluating each architecture in a target task with a standard training procedure, we propose evaluating architectures with a meta-optimized training process (that generates synthetic data in addition to optimizing inner-loop hyper-parameters). We show that doing so significantly reduces the cost of running NAS (Section 3.4).

The goal of these experiments is to find a high-performing CNN architecture for the CIFAR10 image-classification task (Krizhevsky et al., 2009) with limited compute costs. We use the same architecture search-space, training procedure, hyperparameters, and code from Neural Architecture Optimization (Luo et al., 2018), a state-of-the-art NAS method. The search space consists of the topology of two cells: a reduction cell and a convolutional cell. Multiple copies of such cells are stacked according to a predefined blueprint to form a full CNN architecture (see Luo et al. (2018) for more details). The blueprint has two hyperparameters $N$ and $F$ that control how many times the convolutional cell is repeated (depth) and the width of each layer, respectively. Each cell contains $B = 5$ nodes. For each node within a cell, the search algorithm has to choose two inputs as well as two operations to apply to those inputs. The inputs to a node can be previous nodes or the outputs of the last two layers. There are 11 operations to choose from (Appendix C).

Following Luo et al. (2018), we report the performance of our best cell instantiated with $N = 6, F = 36$ after the resulting architecture is trained for a significant amount of time (600 epochs). Since evaluating each architecture in those settings (named *final evaluation* from now on) is time consuming, Luo et al. (2018) uses a surrogate evaluation (named *search evaluation*) to estimate the performance of a given cell wherein a smaller version of the architecture ($N = 3, F = 32$) is trained for less epochs (100) on real data. We further reduce the evaluation time of each cell by replacing the training data in the search evaluation with GTN synthetic data, thus reducing the training time per evaluation by 300x (which we call *GTN evaluation*). While we were able to train GTNs directly on the complex architectures from the NAS search space, training was prohibitively slow. Instead, for these experiments, we optimize our GTN ahead of time using proxy learners described in Appendix A.2, which are smaller fully-convolutional networks (this meta-training took 8h on one p6000 GPU). Interestingly, although we never train our GTN on any NAS architectures, because of generalization, synthetic data from GTNs were still effective for training them.

## 3 RESULTS

We first demonstrate that weight normalization significantly improves the stability of meta-learning, an independent contribution of this paper (Section 3.1). We then show that training with synthetic data is more effective when learning such data jointly with a curriculum that orders its presentation to the learner (Section 3.2). We next show that GTNs can generate a synthetic training set that enables more rapid learning in a few SGD steps than real training data in two supervised learning

domains (MNIST and CIFAR10) and in a reinforcement learning domain (cart-pole, Appendix H). We then apply GTN-synthetic training data for neural architecture search to find high performing architectures for CIFAR10 with limited compute, outperforming comparable methods like weight sharing (Pham et al., 2018) and Graph HyperNetworks (Zhang et al., 2018) (Section 3.4).

We uniformly split the usual MNIST *training* set into training (50k) and validation sets (10k). The training set was used for inner-loop training (for the baseline) and to compute meta-gradients for all the treatments. We used the validation set for hyperparameter tuning and report accuracy on the usual MNIST test set (10k images). We followed the same procedure for CIFAR10, resulting in training, validation, and test sets with 45k, 5k, and 10k examples, respectively. Unless otherwise specified, we ran each experiment 5 times and plot the mean and its 95% confidence intervals from (n=1,000) bootstrapping. Appendix A describes additional experimental details.

## 3.1 Improving Stability with Weight Normalization

To demonstrate the effectiveness of weight normalization for stabilizing and robustifying meta-optimization, we compare the results of running hyperparameter optimization for GTNs with and without weight normalization on MNIST. Figure 1b shows the distribution of the final performance obtained for 20 runs *during* hyperparameter tuning, which reflects how sensitive the algorithms are to hyperparameter settings. Overall, weight normalization substantially improved robustness to hyperparameters and final learner performance, supporting the initial hypothesis.

## 3.2 Improving GTNs with a Curriculum

We experimentally evaluate four different variants of GTNs, each with increasing control over the ordering of the $z$ codes input to the generator, and thus the order of the inputs provided to the learner. The first variant (called *GTN - No Curriculum*), trains a generator to output synthetic training data by sampling the noise vector $z$ for each sample independently from a Gaussian distribution. In the next three GTN variants, the generator is provided with a fixed set of input samples (instead of a noise vector). These input samples are learned along with the generator parameters during GTN training. The second GTN variant (called *GTN - All Shuffled*) learns a fixed set of 4,096 input samples that are presented in a random order without replacement (thus learning controls the data, but not the order in which they are presented). The third variant (called *GTN - Shuffled Batch*) learns 32 batches of 128 samples each (so learning controls which samples coexist within a batch), but the order in which the batches are presented is randomized (without replacement). Finally, the fourth variant (called *GTN - Full Curriculum*) learns a deterministic sequence of 32 batches of 128 samples, giving learning full control. Learning such a curriculum incurs no additional computational expense, as learning the $\mathbf{z}_t$ tensor is computationally negligible and avoids the cost of repeatedly sampling new Gaussian $z$ codes. We plot the test accuracy of a learner (with random initial weights and architecture) as a function of outer-loop iterations for all four variants in Figure 1c. Although *GTNs - No curriculum* can seemingly generate endless data (see Appendix G), it performs worse than the other three variants with a fixed set of generator inputs. Overall, training the GTN with exact ordering of input samples (*GTN - Full Curriculum*) outperforms all other variants.

While curriculum learning usually refers to training on easy tasks first and increasing their difficulty over time, our curriculum goes beyond presenting tasks in a certain order. Specifically, *GTN - Full Curriculum* learns both the order in which to present samples and the specific group of samples to present at the same time. The ability to learn a full curriculum improves GTN performance. For that reason, we adopt that approach for all GTN experiments.

## 3.3 GTNs for Supervised Learning

To explore whether GTNs can generate training data that helps networks learn rapidly, we compare to 3 treatments for MNIST classification. 1) *Real Data* - Training learners with random mini-batches of real data, as is ubiquitous in SGD. 2) *Dataset Distillation* - Training learners with synthetic data, where training examples are directly encoded as tensors optimized by the meta-objective, as in Wang et al. (2019b). 3) *GTN* - Our method where the training data presented to the learner is generated by a neural network. Note that all three methods meta-optimize the inner-loop hyperparameters (i.e. the learning rate and momentum of SGD) as part of the meta-optimization.

We emphasize that producing state-of-the-art (SOTA) performance (e.g. on MNIST or CIFAR) when training with GTN-generated data is *not* important for GTNs. Because the ultimate aim for

GTNs is to accelerate NAS (Section 3.4), what matters is *how well and inexpensively we can identify* architectures that achieve high *asymptotic accuracy* when later trained on the full (real) training set. A means to that end is being able to train architectures rapidly, i.e. with very few SGD steps, because doing so allows NAS to rapidly identify promising architectures. We are thus interested in "few-step accuracy (i.e. accuracy after a few–e.g. 32 or 128–SGD steps). Besides, there are many reasons not to expect SOTA performance with GTNs (Appendix B).

Figure 2a shows that the GTN treatment significantly outperforms the other ones ($p < 0.01$) and trains a learner to be much more accurate when *in the few-step performance regime*. Specifically, for each treatment the figure shows the test performance of a learner following 32 inner-loop training steps with a batch size of 128. We would not expect training on synthetic data to produce higher accuracy than unlimited SGD steps on real data, but here the performance gain comes because GTNs can *compress* the real training data by producing synthetic data that enables learners to learn more quickly than on real data. For example, the original dataset might contain many similar images, where only a few of them would be sufficient for training (and GTN can produce just these few). GTN could also combine many different things that need to be learned about images into one image.

Figure 2b shows the few-step performance of a learner from each treatment after 2000 total outer-loop iterations (~1 hour on a p6000 GPU). For reference, Dataset Distillation (Wang et al., 2019b) reported 79.5% accuracy for a randomly initialized network (using 100 synthetic images vs. our 4,096) and L2T (Fan et al., 2018) reported needing 300x more training iterations to achieve $> 98\%$ MNIST accuracy. Surprisingly, although recognizable as digits and effective for training, GTN-generated images (Figure 2c) were not visually realistic (see Discussion).

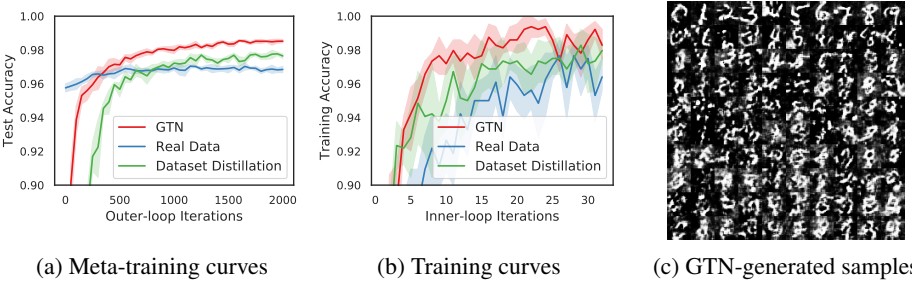

(a) Meta-training curves  (b) Training curves  (c) GTN-generated samples

Figure 2: Teaching MNIST with GTN-generated images. (a) shows MNIST test set few-step accuracy across outer-loop iterations for different sources of inner-loop training data. The inner-loop consists of 32 SGD steps and the outer-loop optimizes MNIST validation accuracy. Our method (GTN) outperforms the two controls (dataset distillation and samples from real data). (b) shows, given final meta-training iteration, how iterations of inner-loop training compare between training data sources (measured by MNIST training set accuracy). (c) shows 100 random samples from the trained GTN. Samples are usually recognizable as digits, but are not realistic (see Discussion). Each column contains samples from a different digit class, and each row is taken from different inner-loop iterations (evenly spaced from the 32 total iterations, with early iterations at the top).

## 3.4 ARCHITECTURE SEARCH WITH GTNs

We next test the benefits of GTN for NAS (GTN-NAS) in CIFAR10, a domain where NAS has previously shown significant improvements over the best architectures produced by armies of human scientists. Figure 3a shows the few-step training accuracy of a learner trained with either GTN-synthetic data or real (CIFAR10) data over meta-training iterations. After 8h of meta-training, training with GTN-generated data was significantly faster than with real data, as in MNIST.

To explore the potential for GTN-NAS to accelerate CIFAR10 architecture search, we investigated the Spearman rank correlation (across architectures sampled from the NAS search space) between accelerated GTN-trained network performance (*GTN evaluation*) and the usual more expensive performance metric used during NAS (*search evaluation*). A correlation plot is shown in Figure 3c; note that a strong correlation implies we can train architectures using GTN evaluation as an inexpensive surrogate. We find that GTN evaluation enables predicting the performance of an architecture efficiently. The rank-correlation between 128 *steps* of training with GTN-synthetic data vs. 100 *epochs* of real data is 0.3606. The correlation improves to 0.5582 when considering the top 50% of archi-

tectures recommended by GTN evaluation scores, which is important because those are the ones that search would select. This improved correlation is slightly stronger than that from *3 epochs* of training with real data (0.5235), a $\sim 9\times$ cost-reduction per trained model.

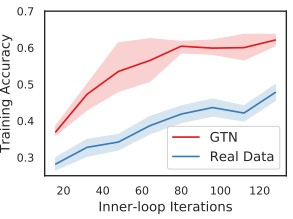
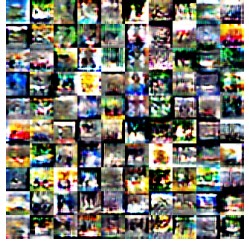
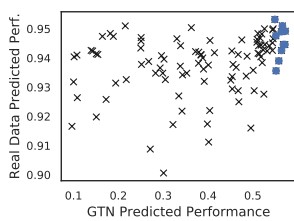

(a) CIFAR10 inner-loop training     (b) CIFAR10 GTN samples     (c) CIFAR10 correlation

Figure 3: Teaching CIFAR10 with GTN-generated images. (a) CIFAR10 training set performance of the final learner (after 1,700 meta-optimization steps) across inner-loop learning iterations. (b) Samples generated by GTN to teach CIFAR10 are unrecognizable, despite being effective for training. Each column contains a different class, and each row is taken from the same inner-loop iteration (evenly spaced from all 128 iterations, early iterations at the top). (c) Correlation between performance prediction using GTN-data vs. Real Data. When considering the top half of architectures (as ranked by GTN evaluation), correlation between GTN evaluation and search evaluation is strong (0.5582 rank-correlation), suggesting that GTN-NAS has potential to uncover high performing architectures at a significantly lower cost. Architectures shown are uniformly sampled from the NAS search space. The top 10% of architectures according to the GTN evaluation (blue squares)– those likely to be selected by GTN-NAS–have high true asymptotic accuracy.

Architecture search methods are composed of several semi-independent components, such as the choice of search space, search algorithm, and proxy evaluation of candidate architectures. GTNs are proposed as an improvement to this last component, i.e. as a new way to quickly evaluate a new architecture. Thus we test our method under the standard search space for CIFAR10, using a simple form of search (random search) for which there are previous benchmark results. In particular, we ran an architecture search experiment where we evaluated 800 randomly generated architectures trained with GTN-synthetic data. We present the performance after *final evaluation* of the best architecture found in Table 1. This experimental setting is similar to that of Zhang et al. (2018). Highlighting the potential of GTNs as an improved proxy evaluation for architectures, we achieve state-of-the-art results when controlling for search algorithm (the choice of which is orthogonal to our contribution). While it is an apples-to-oranges comparison, GTN-NAS is competitive even with methods that use more advanced search techniques than random search to propose architectures (Appendix E). GTN is compatible with such techniques, and would likely improve their performance, an interesting area of future work. Furthermore, because of the NAS search space, the modules GTN found can be used to create even larger networks. A further test of whether GTNs predictions generalize is if such larger networks would continue performing better than architectures generated by the real-data control, similarly scaled. We tried F=128 and show it indeed does perform better (Table 1), suggesting additional gains can be had by searching post-hoc for the correct F and N settings.

## 4 DISCUSSION, FUTURE WORK, AND CONCLUSION

The results presented here suggest potential future applications and extensions of GTNs. Given the ability of GTNs to rapidly train new models, they are particularly useful when training many independent models is required (as we showed for NAS). Another such application would be to teach networks on demand to realize particular trade-offs between e.g. accuracy, inference time, and memory requirements. While to address a range of such trade-offs would ordinarily require training many models ahead of time and selecting amongst them (Elsken et al., 2019), GTNs could instead rapidly train a new network only when a particular trade-off is needed. Similarly, agents with unique combinations of skills could be created on demand when needed.

Interesting questions are raised by the lack of similarity between the synthetic GTN data and real MNIST and CIFAR10 data. That unrealistic and/or unrecognizable images can meaningfully affect NNs is reminiscent of the finding that deep neural networks are easily fooled by unrecognizable

Table 1: Performance of different architecture search methods. Our results report mean $\pm$ SD of 5 evaluations of the same architecture with different initializations. It is common to report scores with and without Cutout (DeVries & Taylor, 2017), a data augmentation technique used during training. We found better architectures compared to other methods that reduce architecture evaluation speed and were tested with random search (Random Search+WS and Random Search+GHN). Increasing the width of the architecture found (F=128) further improves performance. Because each NAS method finds a different architecture, the number of parameters differs. Each method ran once.

| Model | Error(%) | #params | GPU Days |
|---|---|---|---|
| Random Search + GHN (Zhang et al., 2018) | $4.3 \pm 0.1$ | 5.1M | 0.42 |
| Random Search + Weight Sharing (Luo et al., 2018) | 3.92 | 3.9M | 0.25 |
| Random Search + Real Data (baseline) | $3.88 \pm 0.08$ | 12.4M | 10 |
| Random Search + GTN (ours) | $\mathbf{3.84} \pm 0.06$ | 8.2M | 0.67 |
| Random Search + Real Data + Cutout (baseline) | $3.02 \pm 0.03$ | 12.4M | 10 |
| Random Search + GTN + Cutout (ours) | $\mathbf{2.92} \pm 0.06$ | 8.2M | 0.67 |
| Random Search + Real Data + Cutout (F=128) (baseline) | $2.51 \pm 0.13$ | 151.7M | 10 |
| Random Search + GTN + Cutout (F=128) (ours) | $\mathbf{2.42} \pm 0.03$ | 97.9M | 0.67 |

images (Nguyen et al., 2015). It is possible that if neural network architectures were functionally more similar to human brains, GTNs' synthetic data might more resemble real data. However, an alternate (speculative) hypothesis is that the human brain might also be able to rapidly learn an arbitrary skill by being shown unnatural, unrecognizable data (recalling the novel Snow Crash).

The improved stability of training GTNs from weight normalization naturally suggests the hypothesis that weight normalization might similarly stabilize, and thus meaningfully improve, any techniques based on meta-gradients (e.g. MAML (Finn et al., 2017), learned optimizers (Metz et al., 2019), and learned update rules (Metz et al., 2018)). In future work, we will more deeply investigate how consistently, and to what degree, this hypothesis holds.

Both weight sharing and GHNs can be combined with GTNs by using the shared weights or Hyper-Network for initialization of proposed learners and then fine-tuning on GTN-produced data. GTNs could also be combined with more intelligent ways to propose which architecture to sample next such as NAO (Luo et al., 2018). Many other extensions would also be interesting to consider. GTNs could be trained for unsupervised learning, for example by training a useful embedding function. Additionally, they could be used to stabilize GAN training and prevent mode collapse (Appendix I shows encouraging initial results). One particularly promising extension is to introduce a closed-loop curriculum (i.e. one that responds dynamically to the performance of the learner throughout training), which we believe could significantly improve performance. For example, a recurrent GTN that is conditioned on previous learner outputs could adapt its samples to be appropriately easier or more difficult depending on an agent's learning progress, similar in spirit to the approach of a human tutor. Such closed-loop teaching can improve learning (Fan et al., 2018).

An additional interesting direction is having GTNs generate training environments for RL agents. Appendix H shows this works for the simple RL task of CartPole. That could be either for a pre-defined target task, or could be combined with more open-ended algorithms that attempt to continuously generate new, different, interesting tasks that foster learning (Clune, 2019; Wang et al., 2019a). Because GTNs can encode any possible environment, they (or something similar) may be necessary to have truly unconstrained, open-ended algorithms (Stanley et al., 2017). If techniques could be invented to coax GTNs to produce recognizable, human-meaningful training environments, the technique could also produce interesting virtual worlds for us to learn in, play in, or explore.

This paper introduces a new method called Generative Teaching Networks, wherein data generators are trained to produce effective training data through meta-learning. We have shown that such an approach can produce supervised datasets that yield better few-step accuracy than an equivalent amount of real training data, and generalize across architectures and random initializations. We leverage such efficient training data to create a fast NAS method that generates state-of-the-art architectures (controlling for the search algorithm). While GTNs may be of particular interest to the field of architecture search (where the computational cost to evaluate candidate architectures often limits the scope of its application), we believe that GTNs open up an intriguing and challenging line of research into a variety of algorithms that learn to generate their own training data.

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

## APPENDIX A    ADDITIONAL EXPERIMENTAL DETAILS

The outer loop loss function is domain specific. In the supervised experiments on MNIST and CIFAR, the outer loop loss was cross-entropy for logistic regression on real MNIST or CIFAR data. The inner-loop loss matches the outer-loop loss, but with synthetic data instead of real data. Appendix H describes the losses for the RL experiments.

The following equation defines the inner-loop SGD with momentum update for the learner parameters $\theta_t$.

$$\theta_{t+1} = \theta_t - \alpha \sum_{0 \le t' \le t} \beta^{t-t'} \nabla \ell_{\text{inner}}(G(\mathbf{z}_{t'}, \mathbf{y}_{t'}), \mathbf{y}_{t'}, \theta_{t'}), \tag{1}$$

where $\alpha$ and $\beta$ are the learning rate and momentum hyperparameters, respectively. $\mathbf{z}_t$ is a batch of noise vectors that are input to the generator and are sampled from a unit-variance Gaussian. $\mathbf{y}_t$ are a batch of labels for each generated sample/input and are sampled uniformly from all available class labels. Instead of randomly sampling $\mathbf{z}_t$, we can also learn a curriculum by additionally optimizing $\mathbf{z}_t$ directly and keeping $\mathbf{y}_t$ fixed throughout all of training. Results for both approaches (and additional curriculum ablations) are reported in Section 3.2.

### A.1    MNIST EXPERIMENTS:

For the GTN training for MNIST we sampled architectures from a distribution that produces architectures with convolutional (conv) and fully-connectd (FC) layers. All architectures had 2 conv layers, but the number of filters for each layer was sampled uniformly from the ranges $U([32, 128])$ and $U([64, 256])$, respectively. After each conv layer there is a max pooling layer for dimensionality reduction. After the last conv layer, there is a fully-connected layer with number of filters sampled uniformly from the range $U([64, 256])$. We used Kaiming Normal initialization (He et al., 2015) and LeakyReLUs (Maas et al., 2013) (with $\alpha = 0.1$). We use BatchNorm (Ioffe & Szegedy, 2015) for both the generator and the learners. The BatchNorm momentum for the learner was set to 0 (meta-training consistently converged to small values and we saw no significant gain from learning the value).

The generator consisted of 2 FC layers (1024 and $128 * H/4 * H/4$ filters, respectively, where $H$ is the final width of the synthetic image). After the last FC layer there are 2 conv layers. The first conv has 64 filters. The second conv has 1 filter followed by a Tanh. We found it particularly important to normalize (mean of zero and variance of one) all datasets. Hyperparameters are shown in Table 2.

| Hyperparameter | Value |
|---|---|
| Learning Rate | 0.01 |
| Initial LR | 0.02 |
| Initial Momentum | 0.5 |
| Adam Beta_1 | 0.9 |
| Adam Beta_2 | 0.999 |
| Size of latent variable | 128 |
| Inner-Loop Batch Size | 128 |
| Outer-Loop Batch Size | 128 |

Table 2: Hyperparameters for MNIST experiments

### A.2    CIFAR10 EXPERIMENTS:

For GTN training for CIFAR-10, the template architecture is a small learner with 5 convolutional layers followed by a global average pooling and an FC layer. The second and fourth convolution had stride=2 for dimensionality reduction. The number of filters of the first conv layer was sampled uniformly from the range $U([32, 128])$ while all others were sampled uniformly from the range $U([64, 256])$. Other details including the generator architecture were the same as the MNIST experiments, except the CIFAR generator's second conv layer had 3 filters instead of 1. Hyperparameters

used can be found in Table 3. For CIFAR10 we augmented the real training set when training GTNs with random crops and horizontal flips. We do not add weight normalization to the final architectures found during architecture search, but we do so when we train architectures with GTN-generated data during architecture search to provide an estimate of their asymptotic performance.

| Hyperparameter | Value |
|---|---|
| Learning Rate | 0.002 |
| Initial LR | 0.02 |
| Initial Momentum | 0.5 |
| Adam Beta_1 | 0.9 |
| Adam Beta_2 | 0.9 |
| Adam $\epsilon$ | 1e-5 |
| Size of latent variable | 128 |
| Inner-loop Batch Size | 128 |
| Outer-loop Batch Size | 256 |

Table 3: Hyperparameters for CIFAR10 experiments

## APPENDIX B  REASONS GTNS ARE NOT EXPECTED TO PRODUCE SOTA ACCURACY VS. ASYMPTOTIC PERFORMANCE WHEN TRAINING ON REAL DATA

There are three reasons not to expect SOTA accuracy levels for the learners trained on synthetic data: (1) we train for very few SGD steps (32 or 128 vs. tens of thousands), (2) SOTA performance results from architectures explicitly designed (with much human effort) to achieve record accuracy, whereas GTN produces compressed training data optimized to generalize across diverse architectures with the aim of quickly evaluating a new architecture's potential, and (3) SOTA methods often use data outside of the benchmark dataset and complex data-augmentation schemes.

## APPENDIX C  CELL SEARCH SPACE

When searching for the operations in a CNN cell, the 11 possible operations are listed below.

- identity
- $1 \times 1$ convolution
- $3 \times 3$ convolution
- $1 \times 3 + 3 \times 1$ convolution
- $1 \times 7 + 7 \times 1$ convolution
- $2 \times 2$ max pooling
- $3 \times 3$ max pooling
- $5 \times 5$ max pooling
- $2 \times 2$ average pooling
- $3 \times 3$ average pooling
- $5 \times 5$ average pooling

## APPENDIX D  COMPUTATION AND MEMORY COMPLEXITY

With the traditional training of DNNs with back-propagation, the memory requirements are proportional to the size of the network because activations during the forward propagation have to be stored for the backward propagation step. With meta-gradients, the memory requirement also grows with the number of inner-loop steps because all activations and weights have to be stored for the

2nd order gradient to be computed. This becomes impractical for large networks and/or many inner-loop steps. To reduce the memory requirements, we utilize gradient-checkpointing (Griewank & Walther, 2000) by only storing the computed weights of learner after each inner-loop step and re-computing the activations during the backward pass. This trick allows us to compute meta-gradients for networks with 10s of millions of parameters over hundreds of inner-loop steps in a single GPU. While in theory the computational cost of computing meta-gradients with gradient-checkpointing is 4x larger than computing gradients (and 12x larger than forward propagation), in our experiments it is about 2.5x slower than gradients through backpropagation due to parallelism. We could further reduce the memory requirements by utilizing reversable hypergradients (Maclaurin et al., 2015), but, in our case, we were not constrained by the number of inner-loop steps we can store in memory.

## APPENDIX E    EXTENDED NAS RESULTS

In the limited computation regime (less than 1 day of computation), the best methods were, in order, GHN, ENAS, GTN, and NAONet with a mean error of 2.84%, 2.89%, 2.92%, and 2.93%, respectively. A 0.08% difference on CIFAR10 represents 8 out of the 10k test samples. For that reason, we consider all of these methods as state of the art. Note that out of the four, GTN is the only one relying on Random Search for architecture proposal.

Table 4: Performance of different architecture search methods. Search with our method required 16h total, of which 8h were spent training the GTN and 8h were spent evaluating 800 architectures with GTN-produced synthetic data. Our results report mean $\pm$ SD of 5 evaluations of the same architecture with different initializations. It is common to report scores with and without Cutout (DeVries & Taylor, 2017), a data augmentation technique used during training. We found better architectures compared to other methods using random search (Random-WS and GHN-Top) and are competitive with algorithms that benefit from more advanced search methods (e.g. NAONet and ENAS employ non-random architecture proposals for performance gains; GTNs could be combined with such non-random proposals, which would likely further improve performance). Increasing the width of the architecture found (F=128) further improves performance.

| Model | Error(%) | #params | Random | GPU Days |
|---|---|---|---|---|
| NASNet-A (Zoph & Le, 2017) | 3.41 | 3.3M | ✗ | 2000 |
| AmoebaNet-B + Cutout (Real et al., 2019) | 2.13 | 34.9M | ✗ | 3150 |
| DARTS + Cutout (Liu et al., 2018b) | 2.83 | 4.6M | ✗ | 4 |
| NAONet + Cutout (Luo et al., 2018) | 2.48 | 10.6M | ✗ | 200 |
| NAONet-WS (Luo et al., 2018) | 3.53 | 2.5M | ✗ | 0.3 |
| NAONet-WS + Cutout (Luo et al., 2018) | 2.93 | 2.5M | ✗ | 0.3 |
| ENAS (Pham et al., 2018) | 3.54 | 4.6M | ✗ | 0.45 |
| ENAS + Cutout (Pham et al., 2018) | 2.89 | 4.6M | ✗ | 0.45 |
| GHN Top-Best + Cutout (Zhang et al., 2018) | 2.84 $\pm$ 0.07 | 5.7M | ✗ | 0.84 |
| GHN Top (Zhang et al., 2018) | 4.3 $\pm$ 0.1 | 5.1M | ✓ | 0.42 |
| Random-WS (Luo et al., 2018) | 3.92 | 3.9M | ✓ | 0.25 |
| Random Search + Real Data (baseline) | 3.88 $\pm$ 0.08 | 12.4M | ✓ | 10 |
| RS + Real Data + Cutout (baseline) | 3.02 $\pm$ 0.03 | 12.4M | ✓ | 10 |
| RS + Real Data + Cutout (F=128) (baseline) | 2.51 $\pm$ 0.13 | 151.7M | ✓ | 10 |
| Random Search + GTN (ours) | **3.84** $\pm$ 0.06 | 8.2M | ✓ | 0.67 |
| Random Search + GTN + Cutout (ours) | **2.92** $\pm$ 0.06 | 8.2M | ✓ | 0.67 |
| RS + GTN + Cutout (F=128) (ours) | **2.42** $\pm$ 0.03 | 97.9M | ✓ | 0.67 |

## APPENDIX F    CONDITIONED GENERATOR VS. XY-GENERATOR

Our experiments in the main paper conditioned the generator to create data with given labels, by concatenating a one-hot encoded label to the input vector. We also explored an alternative approach where the generator itself produced a target probability distribution to label the data it generates. Because more information is encoded into a soft label than a one-hot encoded one, we expected an improved training set to be generated by this variant. Indeed, such a "dark knowledge" distillation setup has been shown to perform better than learning from labels (Hinton et al., 2015).

However, the results in Figure 4 indicate that jointly generating both images and their soft labels under-performs generating only images, although the result could change with different hyperparameter values and/or innovations that improve the stability of training.

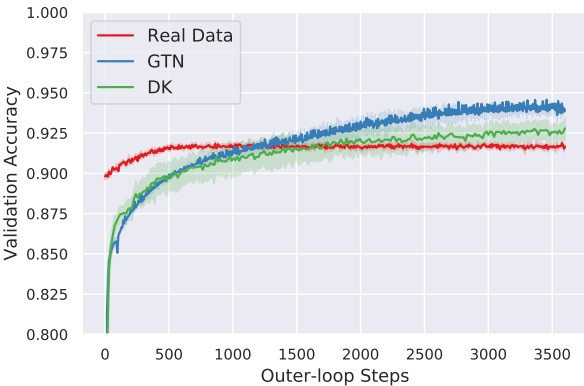

Figure 4: Comparison between a conditional generator and a generator that outputs an image/label pair. We expected the latter "dark knowledge" approach to outperform the conditional generator, but that does not seem to be the case. Because initialization and training of the dark knowledge variant were more sensitive, we believe a more rigorous tuning of the process could lead to a different result.

## APPENDIX G    GTN GENERATES (SEEMINGLY) ENDLESS DATA

While optimizing images directly (i.e. optimizing a fixed tensor of images) would result in a fixed number of samples, optimizing a generator can potentially result in an unlimited amount of new samples. We tested this generative capability by generating more data during evaluation (i.e. with no change to the meta-optimization procedure) in two ways. In the first experiment, we increase the amount of data in each inner-loop optimization step by increasing the batch size (which results in lower variance gradients). In the second experiment, we keep the number of samples per batch fixed, but increase the number of inner-loop optimization steps for which a new network is trained. Both cases result in an increased amount of training data. If the GTN generator has overfit to the number of inner-loop optimization steps during meta-training and/or the batch size, then we would not expect performance to improve when we have the generator produce more data. However, an alternate hypothesis is that the GTN is producing a healthy distribution of training data, irrespective of exactly how it is being used. Such a hypothesis would be supported by performance increase in these experiments.

Figure 5a shows performance as a function of increasing batch size (beyond the batch size used during meta-optimization, i.e. 128). The increase in performance of GTN means that we can sample larger training sets from our generator (with diminishing returns) and that we are not limited by the choice of batch size during training (which is constrained due to both memory and computation requirements).

Figure 5b shows the results of generating more data by increasing the number of inner-loop optimization steps. Generalization to more inner-loop optimization steps is important when the number of inner-loop optimization steps used during meta-optimization is not enough to achieve maximum performance. This experiment also tests the generalization of the optimizer hyperparameters because they were optimized to maximize learner performance after a fixed number of steps. There is an increase in performance of the learner trained on GTN-generated data as the number of inner-loop optimization steps is increased, demonstrating that the GTN is producing generally useful data instead of overfitting to the number of inner-loop optimization steps during training (Figure 5b). Extending the conclusion from Figure 2b, in the very low data regime, GTN is significantly better than training on real data ($p < 0.05$). However, as more inner-loop optimization steps are taken and thus more unique data is available to the learner, training on the real data becomes more effective than learning from synthetic data ($p < 0.05$) (see Figure 5b).

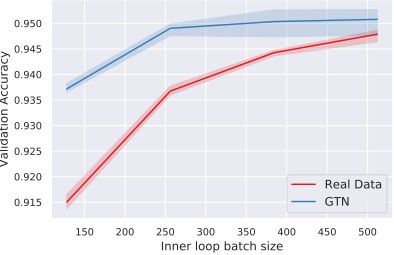 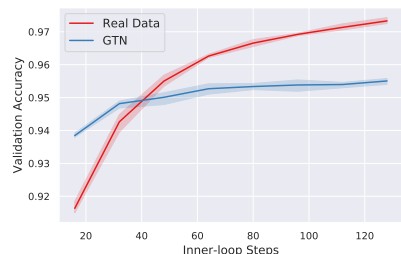

(a) Increasing inner-loop batch size         (b) Increasing inner-loop optimization steps

Figure 5: (a) The left figure shows that even though GTN was meta-trained to generate synthetic data of batch size 128, sampling increasingly larger batches results in improved learner performance (the inner-loop optimization steps are fixed to 16). (b) The right figure shows that increasing the number of inner-loop optimization steps (beyond the 16 steps used during meta-training) improves learner performance. The performance gain with real data is larger in this setting. This improvement shows that GTNs do not overfit to a specific number of inner-loop optimization steps.

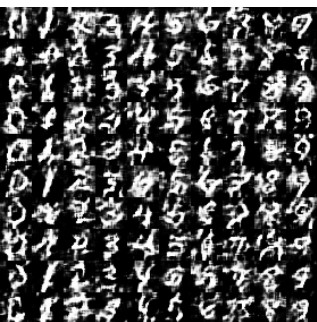

Figure 6: GTN samples w/o curriculum.

Another interesting test for our generative model is to test the distribution of learners after they have trained on the synthetic data. We want to know, for instance, if training on synthetic samples from one GTN results in a functionally similar set of learner weights regardless of learner initialization (this phenomena can be called learner mode collapse). Learner mode collapse would prevent the performance gains that can be achieved through ensembling diverse learners. We tested for learner mode collapse by evaluating the performance (on held-out data and held-out architecture) of an ensemble of 32 randomly initialized learners that are trained on independent batches from the *same GTN*. To construct the ensemble, we average the predicted probability distributions across the learners to compute a combined prediction and accuracy. The results of this experiment can be seen in Figure 7, which shows that the combined performance of an ensemble is better (on average) than an individual learner, providing additional evidence that the distribution of synthetic data is healthy and allows ensembles to be harnessed to improve performance, as is standard with networks trained on real data.

## APPENDIX H    GTN FOR RL

To demonstrate the potential of GTNs for RL, we tested our approach with a small experiment on the classic CartPole test problem (see Brockman et al. (2016) for details on the domain). We conducted this experiment before the discovery that weight normalization improves GTN training, so these experiments do not feature it; it might further improve performance. For this experiment, the meta-objective the GTN is trained with is the advantage actor-critic formulation: $\log \pi(a|\theta_\pi)(R - V(s; \theta_v))$ (Mnih et al., 2016). The state-value $V$ is provided by a separate neural network trained to estimate the average state-value for the learners produced so far during meta-training. The learners train on synthetic data via a single-step of SGD with a batch size of 512 and a mean squared error

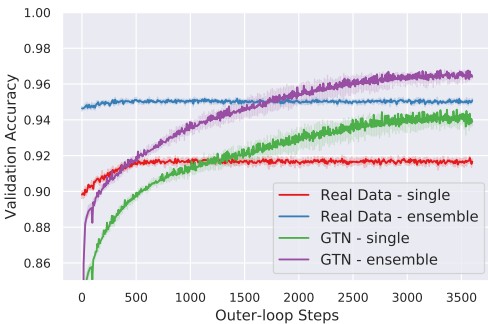

Figure 7: Performance of an ensemble of GTN learners vs. individual GTN learners. Ensembling a set of neural networks that each had different weight initializations, but were trained on data from the *same GTN* substantially improves performance. This result provides more evidence that GTNs generate a healthy distribution of training data and are not somehow forcing the learners to all learn a functionally equivalent solution.

regression loss, meaning the inner loop is supervised learning. The outer-loop is reinforced because the simulator is non-differentiable. We could have also used an RL algorithm in the inner loop. In that scenario the GTN would have to learn to produce an entire synthetic world an RL agent would learn in. Thus, it would create the initial state and then iteratively receive actions and generate the next state and optionally a reward. For example, a GTN could learn to produce an entire MDP that an agent trains on, with the meta-objective being that the trained agent then performs well on a target task. We consider such synthetic (PO)MDPs an exciting direction for future research.

The score on CartPole is the number of frames out of 200 for which the pole is elevated. Both GTN and an A2C (Mnih et al., 2016) control effectively solve the problem (Figure 8). Interestingly, training GTNs takes the same number of simulator steps as training a single learner with policy-gradients (Figure 8). Incredibly, however, once trained, the synthetic data from a GTN can be used to train a learner to maximum performance in a single SGD step! While that is unlikely to be true for harder target RL tasks, these results suggest that the speed-up for architecture search from using GTNs in the RL domain can be even greater than in supervised domain.

The CartPole experiments feature a single-layer neural network with 64 hidden units and a tanh activation function for both the policy and the value network. The inner-loop batch size was 512 and the number of inner-loop training iterations was 1. The observation space of this environment consists of a real-valued vector of size 4 (Cart position, Cart velocity, Pole position, Pole velocity). The action space consists of 2 discrete actions (move left or move right). The outer loop loss is the reward function for the target domain (here, pole-balancing). The inner loop loss is mean squared error (i.e. the network is doing supervised learning on the state-action mapping pairs provided by the GTN).

## APPENDIX I  SOLVING MODE COLLAPSE IN GANS WITH GTNS

We created an implementation of generative adversarial networks (GANs) (Goodfellow et al., 2014) and found they tend to generate the same class of images (e.g. only 1s, Figure 9), which is a common training pathology in GANs known as mode collapse (Srivastava et al., 2017). While there are techniques to prevent mode collapse (e.g. minibatch discrimination and historical averaging (Salimans et al., 2016)), we hypothesized that combining the ideas behind GTNs and GANs might provide a different, additional technique to help combat mode collapse. The idea is to add a discriminator to the GTN forcing the data it generates to both be realistic and help a learner perform well on the meta-objective of classifying MNIST. The reason this approach should help prevent mode collapse is that if the generator only produces one class of images, a learner trained on that data will not be able to classify all classes of images. This algorithm (GTN-GAN) was able to produce realistic images with no identifiable mode collapse (Figure 10). GTNs offer a different type of solution to the issue of mode collapse than the many that have been proposed, adding a new tool to our toolbox

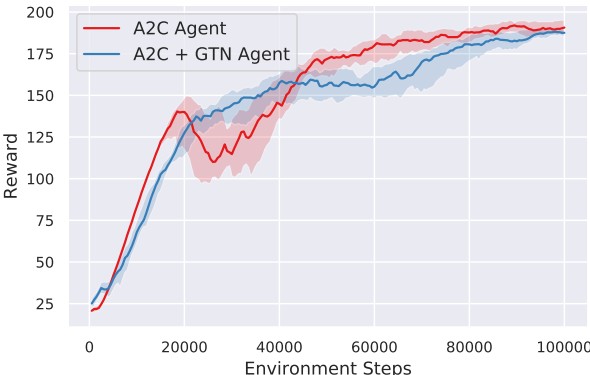

Figure 8: An A2C Agent control trains a single policy throughout all of training, while the GTN method starts with a new, randomly initialized network *at each iteration* and produces the plotted performance *after a single step of SGD*. This plot is difficult to parse because of that difference: it compares the accumulated performance of A2C across all environment steps up to that point vs. the performance achieved with GTN data *in a single step of SGD from a single batch of synthetic data*. Thus, at the 100,000[th] step of training, GTNs enable training a newly initialized network to the given performance (of around 190) 100,000 times faster with GTN synthetic data than with A2C from scratch. With GTNs, we can therefore train many new, high-performing agents quickly. That would be useful in many ways, such as greatly accelerating architecture search algorithms for RL. Of course, these results are on a simple problem, and (unlike our supervised learning experiments) have not yet shown that the GTN data works with different architectures, but these results demonstrate the intriguing potential of GTNs for RL. One reason we might expect even larger speedups for RL vs. supervised learning is because a major reason RL is sample inefficient is because it requires exploration to figure out how to solve the problem. However, once that exploration has been done, the GTN can produce data to efficiently teach that solution to a new architecture. RL thus represents an exciting area of future research for GTNs. Performing that research is beyond the scope of this paper, but we highlight the intriguing potential here to inspire such future work.

for solving that problem. Note we do not claim this approach is better than other techniques to prevent mode collapse, only that it is an interesting new type of option, perhaps one that could be productively combined with other techniques.

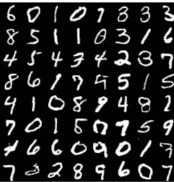 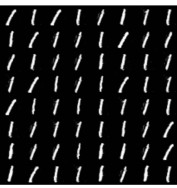

Figure 9: **Images generated by a basic GAN on MNIST before and after mode collapse.** The left image shows GAN-produced images early in GAN training and the right image shows GAN samples later in training after mode collapse has occurred due to training instabilities.

## APPENDIX J    ADDITIONAL MOTIVATION

There is an additional motivation for GTNs that involves long-term, ambitious research goals: GTN is a step towards algorithms that generate their own training environments, such that agents trained in them eventually solve tasks we otherwise do not know how to train agents to solve (Clune, 2019). It is important to pursue such algorithms because our capacity to conceive of effective training environments on our own as humans is limited, yet for our learning algorithms to achieve their full potential they will ultimately need to consume vast and complex curricula of learning challenges

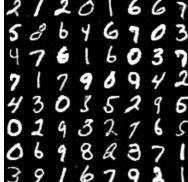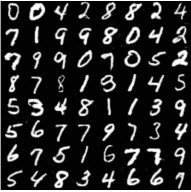

Figure 10: **Images generated by a GTN with an auxiliary GAN loss.** Combining GTNs with GANs produces far more realistic images than GTNs alone (which produced alien, unrecognizable images, Figure 6). The combination also stabilizes GAN training, preventing mode collapse.

and data. Algorithms for generating curricula, such as the the paired open-ended trailblazer (POET) algorithm (Wang et al., 2019a), have proven effective for achieving behaviors that would otherwise be out of reach, but no algorithm yet can generate completely unconstrained training conditions. For example, POET searches for training environments within a highly restricted preconceived space of problems. GTNs are exciting because they can encode a rich set of possible environments with minimal assumptions, ranging from labeled data for supervised learning to (in theory) entire complex virtual RL domains (with their own learned internal physics). Because RNNs are Turing-complete (Siegelmann & Sontag, 1995), GTNs should be able to theoretically encode all possible learning environments. Of course, while what is theoretically possible is different from what is achievable in practice, GTNs give us an expressive environmental encoding to begin exploring what potential is unlocked when we can learn to generate sophisticated learning environments. The initial results presented here show that GTNs can be trained end-to-end with gradient descent through the entire learning process; such end-to-end learning has proven highly scalable before, and may similarly in the future enable learning expressive GTNs that encode complex learning environments.

