# OpenReview forum: "Generative Teaching Networks: Accelerating Neural Architecture Search by Learning  to Generate Synthetic Training Data"
_ICLR.cc/2020/Conference — Reject_

### Official Review · AnonReviewer3 · 2019-10-24
**Official Blind Review #3**

**Rating:** 6

**Review:**

This paper proposes an algorithm for generating training data to help other machine learning agents learn faster. The proposed Generative Teaching Networks (GTNs) are networks that are trained to generate training data for other networks and are trained jointly with these other networks by back propagating through the entire learning problems via meta-gradients. They also show how weight normalization can help stabilize the training of GTNs. The paper is well-written overall and easy to follow, except for a few typos that can be fixed for the camera ready. The main idea of the paper is quite simple and it’s nice to see it works well. I’m actually surprised it has not been proposed before, but I am also not very familiar with this research area. For these reasons, I lean towards accepting this paper, although I have a few comments that I would like to see addressed for the camera ready version.

The authors present experiments where they apply GTNs on image classification and neural architecture search. GTN does indeed seem to do better than the baselines for these problems. However, for MNIST and CIFAR it looks like the models being used may not be that good, as it’s quite easy to obtain better performance than the results shown in the paper. I would be curious to know why the authors did not use a better architecture and also what the architecture they used actually is. I am unfortunately not familiar with neural architecture search to be able to evaluate their experimental setup in that case.

Regarding the curriculum used in Section 3.2, my understanding is that the full curriculum approach has an additional computational cost vs the no curriculum approach, as you have to learn that curriculum. Thus, even though the result the authors present is interesting and verifies that GTNs learn a useful curriculum, I would also like to see curves of how accuracy improves per computational unit (e.g., the horizontal axis could be CPU training time). This would allow us to see whether learning a curriculum this way is in fact practically useful. It may just as well be that it is too expensive and training without it is faster.

The authors show example images generated by GTNs and, as they also mention, these images do not look very realistic. It would be good to have some explanation/analysis around this. Could it be that these are images that are “hard” for the classifier? (e.g., thinking in terms of support vector machines, do these images lie in or close to the margin of the classifier?). I would love to seem an analysis around this and a couple of proposed explanations.

I would also like to see more details on the actual architecture used for the experiments as I feel that the paper does not provide enough information to reproduce the results.

**Experience Assessment:**

I have read many papers in this area.

**Review Assessment: Checking Correctness Of Derivations And Theory:**

N/A

**Review Assessment: Checking Correctness Of Experiments:**

I assessed the sensibility of the experiments.

**Review Assessment: Thoroughness In Paper Reading:**

I read the paper at least twice and used my best judgement in assessing the paper.

---

> ### Author Response · Authors · 2019-11-11
> **Response to review #3**
>
> Thank you for your valuable and constructive feedback. We are glad you think our paper is well written. We will make all suggested changes to improve it.
>
> > The authors present experiments where they apply GTNs on image classification and neural architecture search. GTN does indeed seem to do better than the baselines for these problems. However, for MNIST and CIFAR it looks like the models being used may not be that good, as it’s quite easy to obtain better performance than the results shown in the paper. I would be curious to know why the authors did not use a better architecture and also what the architecture they used actually is. I am unfortunately not familiar with neural architecture search to be able to evaluate their experimental setup in that case.
>
> Please see our #mainIssue response in the reply to all reviewers.
>
> > Regarding the curriculum used in Section 3.2, my understanding is that the full curriculum approach has an additional computational cost vs the no curriculum approach, as you have to learn that curriculum. Thus, even though the result the authors present is interesting and verifies that GTNs learn a useful curriculum, I would also like to see curves of how accuracy improves per computational unit (e.g., the horizontal axis could be CPU training time). This would allow us to see whether learning a curriculum this way is in fact practically useful. It may just as well be that it is too expensive and training without it is faster.
>
> The curriculum version does not have any additional cost because the curriculum is learned concurrently with the rest of the system (in fact, it’s slightly cheaper to train, as described next). The only difference (during meta-training and meta-testing) is that in the no-curriculum version we sample a Z code from a Gaussian and pass that as input to the GTN and, in the curriculum version, we directly train a series of Z codes that are sequentially passed to the GTN over the N inner-loop steps. The extra compute to update that Z-code block is negligible. We will clarify in the paper that a single meta-iteration with GTN costs virtually the same amount of computation regardless of the curriculum-type (as does inference).
>
> > The authors show example images generated by GTNs and, as they also mention, these images do not look very realistic. It would be good to have some explanation/analysis around this. Could it be that these are images that are “hard” for the classifier? (e.g., thinking in terms of support vector machines, do these images lie in or close to the margin of the classifier?). I would love to seem an analysis around this and a couple of proposed explanations.
>
> You raise an interesting hypothesis that we have considered (amongst many). We have some ideas for experiments we can do to try to shed light on this issue broadly. We will actively begin them and report back within the discussion window. One possibility (if none of our experiments are conclusive) is that in the discussion we could list all of the different hypotheses we have come up with that might explain this phenomenon and say that investigating them is an interesting area for future work. Would you like that approach?
>
> > I would also like to see more details on the actual architecture used for the experiments as I feel that the paper does not provide enough information to reproduce the results.
>
> All architectures used are described in SI Appendix A, B, and C. We will update the main text to better emphasize where these details are located. We will also release our code and trained models to ensure reproducible results.
>
> Thank you again for your valuable and constructive feedback. Please let us know if there are any additional changes you would like to see made.

---

> > ### Comment · AnonReviewer3 · 2019-11-15
> > **Response to Rebuttal**
> >
> > Thank you for the response and the explanations provided! Your responses did clarify a few of my questions and it is good you are also planning to address them in the paper. Regarding the images not looking very realistic, yes I would love to see some plausible hypotheses along with a brief analysis on whether they may be valid. I also believe this would be interesting for readers as this question is likely to pop up in the minds while reading your paper. Regarding the curriculum cost, I find your response interesting and would like to see it included in the paper. Having said that, my score remains the same as I believe it's a fair evaluation of this paper. Thanks again for all clarifications!

---

> > > ### Author Response · Authors · 2019-11-15
> > > **Re: Response to Rebuttal**
> > >
> > > Thank you. We have already included the changes in response to your original comments in the paper, including regarding that the curriculum does not incur additional cost. We will include a new section in SI that lists our hypotheses for why the images look unrealistic. We have promising new experimental results that shed some light on this question, but they (and their description) will not be ready to post before the end of the rebuttal window in a few hours (after which we are not able to upload a new PDF).  If there is any chance that their inclusion would cause you to further improve your score, please let us know and we can post a preliminary version of these analyses in the next few hours. We would then substitute in finalized versions for them in the final version of the paper. Either way, thank you again for your time and insightful input, and for voting to accept the paper.

---

### Official Review · AnonReviewer1 · 2019-10-24
**Official Blind Review #1**

**Rating:** 6

**Review:**

Summary:
The paper proposes Generative Teaching Networks, which aims to generate synthetic training data
for a given prediction problem. The authors demonstrate its use in an MNIST prediction task
and a neural architecture search task on Cifar10.
I do not find the idea compelling nor the empirical idea convincing enough to warrant acceptance at
ICLR.


Detailed Comments:

At a high level, the motivation for data generation in order to improve a given prediction problem
is not clear. From a statistical perspective, one can only do so well given a certain amount of
training data, and being able to generate new data would suggest that one can do arbitrarily better
by simply creating more data -- this is not true.

While data augmentation techniques have improved accuracy in many cases, they have also relied
heavily on domain knowledge about the problem, such as mirroring, cropping for images. The proposed
GTN model does not seem to incorporate such priors and I would be surprised that one can do better
with such synthetically generated data.
Indeed, the proposed approach does not do better than the best performing models on MNIST.

The authors use GTNs in a NAS problem where they use the accuracy on the generated images as a proxy
for the validation accuracy. As figure 4c illustrates there actually does not seem to be much
correlation between the accuracies on the synthetic and real datasets.
While Table 1 indicates that they outperform some baselines, I do not find them compelling. This
could simply be because random search is a coarse optimization method (and hence the proposed metric
may not do well on more sophisticated search techniques).
    - On a side note, why is evaluating on the synthetic images cheaper than evaluating on the
      original images?
    - What is the rank-correlation metric used? Did you try more standard correlation metrics such
      as Pearson's coefficient?


=================
Post rebuttal

Having read the rebuttal, the comments from other reviewers, and the updated manuscript, I am more positive about the paper now. I agree that with reviewer 2 that the proposed approach is interesting and could be a method to speed up NAS in new domains. I have upgraded my score to reflect this.

My only remaining issue is that the authors should have demonstrated this on new datasets (by running other methods on these datasets) instead of sticking to the same old datasets. However, this is the standard practice in the NAS literature today.

**Experience Assessment:**

I do not know much about this area.

**Review Assessment: Checking Correctness Of Derivations And Theory:**

N/A

**Review Assessment: Checking Correctness Of Experiments:**

I assessed the sensibility of the experiments.

**Review Assessment: Thoroughness In Paper Reading:**

I made a quick assessment of this paper.

---

> ### Author Response · Authors · 2019-11-11
> **Response to Review #1 (part 2)**
>
> > On a side note, why is evaluating on the synthetic images cheaper than evaluating on the
>       original images?
>
> Image for image, they are equally fast. What our results show, however, is that because the GTN-produced data is trained to lead to fast learning, we need to train on far less of it to achieve the same performance as real training. Specifically, we can achieve equal performance with ~4x fewer SGD steps by using GTN-data vs. real data (Figure 4a).
>
> More importantly, our approach is trying to, as quickly as possible, estimate the asymptotic performance of an architecture when trained on real data. To do so, we train on GTN-data for a very low number of SGD steps (128). We show that GTNs produces an equally accurate rank-correlation in architectures (i.e. using the proxy to estimate their asymptotic-performance ordering from best to worst) just as well as when training with real data, but using 9x fewer SGD steps. That provides significant compute savings. On our development computer, for example, it takes about 30 seconds per architecture for our approach vs. 270 seconds with real data.
>
> > What is the rank-correlation metric used? Did you try more standard correlation metrics such as Pearson's coefficient?
>
> We use Spearman’s rank-correlation because NAS needs only to know the ranks of the networks, not their absolute performance. Spearman’s rank-correlation is in effect the rank-based version of Pearson’s coefficient.

---

> ### Author Response · Authors · 2019-11-11
> **Response to review #1 (part 1)**
>
> Thank you for your feedback, we hope we can address your concerns and clear up any misunderstandings.
>
> > At a high level, the motivation for data generation in order to improve a given prediction problem is not clear. From a statistical perspective, one can only do so well given a certain amount of training data, and being able to generate new data would suggest that one can do arbitrarily better by simply creating more data -- this is not true.
>
> We agree. This is similar to the “extra juice” comment from reviewer 2, and we address it in our #mainIssue response above.
>
> > While data augmentation techniques have improved accuracy in many cases, they have also relied heavily on domain knowledge about the problem, such as mirroring, cropping for images. The proposed GTN model does not seem to incorporate such priors and I would be surprised that one can do better with such synthetically generated data. Indeed, the proposed approach does not do better than the best performing models on MNIST.
>
> Please see our #mainIssue response to all reviewers.
>
> > The authors use GTNs in a NAS problem where they use the accuracy on the generated images as a proxy for the validation accuracy. As figure 4c illustrates there actually does not seem to be much correlation between the accuracies on the synthetic and real datasets.
>
> Figure 4c indicates that there is a correlation between the 2 metrics, although it is weak. More importantly, there is a much stronger rank-correlation among the high performing architectures. Many NAS papers use the performance after training for a short period of time as a proxy for asymptotic performance. Our paper shows that training with GTN data is equivalent to training with real data for 9x longer, providing a significant speedup over the classic (i.e. real-data) approach to short-training. Finally, GTN-NAS does find architectures that are competitive with the SOTA-NAS methods, demonstrating that it is in fact able to provide good-enough estimates to yield a SOTA-competitive result. Given that, it seems fair to say that the correlation, whatever its numerical value, is functionally useful.
>
> > While Table 1 indicates that they outperform some baselines, I do not find them compelling. This could simply be because random search is a coarse optimization method (and hence the proposed metric may not do well on more sophisticated search techniques).
>
> Our technique provides a drop-in replacement for training with real data. For example, a common method is to train on real data for N steps of SGD and use the resulting performance as a proxy for asymptotic performance. This allows the NAS algorithm to evaluate many architectures quickly. Then different NAS algorithms do different things with that proxy. Our paper shows that GTN can provide an even faster (and thus computationally cheaper) proxy. Arguably the simplest algorithm that can use this proxy is Random Search. In it, T architectures are generated at random, the proxy value for each is calculated, the best architecture according to the proxy is selected, and that architecture is then trained on real data until convergence (with this asymptotic performance being the final score for the algorithm). We chose this as our algorithm because it is so simple that it isolates the speedup effects of GTN vs. Real Data. The point is not that beating RS-NAS is itself a victory because, as you say, there are better methods than RS-NAS. Instead, the point is that this clear experiment shows the benefits of GTN-speedups vs. Real Data without any complex, confounding algorithmic complexities.
>
> While it is of course possible that our algorithm would not work well with more sophisticated search techniques that use proxies based on short-training with real-data, we have no reason to believe that is true. We think testing this hypothesis is an interesting area for future work, but is not required to demonstrate the benefits of GTN-data vs. Real Data **when holding all else equal**.
>
> All of that said, even when compared against more sophisticated search techniques (Table 4, SI) we remain competitive against the other NAS methods that use limited computation (i.e. less than 1 GPU day). Excitingly, adding GTN data into those algorithms should thus further improve them, as we mention in the paper.

---

### Official Review · AnonReviewer2 · 2019-10-26
**Official Blind Review #2**

**Rating:** 3

**Review:**

This paper proposes a meta-learning algorithm Generative Teaching Networks (GTN) to generate fake training data for models to learn more accurate models. In the inner loop, a generator produces training data and the learner takes gradient steps on this data. In the outer loop, the parameters of the generator are updated by evaluating the learner on real data and differentiating through the gradient steps of the inner loop. The main claim is this method is shown to give improvements in performance on supervised learning for MNIST and CIFAR10. They also suggest weight normalization patches up instability issues with meta-learning and evaluate this in the supervised learning setting, and curriculum learning for GTNs.

To me, the main claim is very surprising and counter-intuitive - it is not clear where the extra juice is coming from, as the algorithm does not assume any extra information. The actual results I believe do not bear out this claim because the actual results on MNIST and CIFAR10 are significantly below state of the art. On MNIST, GTN achieves about 98% accuracy and the baseline “Real Data” achieves  <97% accuracy, while the state of the art is about 99.7% and well-tuned convnets without any pre-processing or fancy extras achieve about 99% according to Yann LeCunn’s website. The disparity on CIFAR seems to be less egregious but the state of the art stands at 99% while the best GTN model (without cutout) achieves about 96.2% which matches good convnets and is slightly worse than neural architecture search according to https://paperswithcode.com/sota/image-classification-on-cifar-10.

This does not negate the potential of GTNs which I feel are an interesting approach, but I believe the paper should be more straightforward with the presentation of these results. The current results basically  show that GTNs improve the performance of learners with bad hyper-parameters. On problems that are not as well-studied as MNIST or CIFAR10 this could still be very valuable (as we do not know what performance is good or bad in advance). Based on the results, GTN does seem to be a significant step forward in synthetic data generation for learning compared to prior work (Zhang 2018, Luo 2018).

The paper proposes two other contributions: using weight normalization for meta-learning and curriculum learning for GTNs. Weight normalization is shown to stabilize GTNs on MNIST. I think the paper oversteps in the relevant method section, hypothesizing it may stabilize meta-learning more broadly. The paper should present a wider set of experiments to make this claim convincing. But the point for GTNs on MNIST nevertheless stands. For curriculum learning: the description of the method is done across section 2 and section 3.2 and does not really describe it completely. How exactly are the samples chosen in GTN - All Shuffled? How does GTN - Full Curriculum and Shuffled Batch parametrize the order of the samples so that it can be learned? I suggest that this information is all included as a subsection in the method (section 2). The results seem to show the learned curriculum is superior to no curriculum.

At a high level it would be very surprising to me if the way forward for better discriminative models was to learn good generative models and use them again for training discriminative models, simply because discriminative models have proved thus far significantly easier to train. If this work does eventually show this result, it would be a very interesting result. At the moment, I believe it does not, but I would be happy to change my mind if the authors provide convincing evidence. Alternatively, I feel that the paper could be a valuable contribution to the community if the writing is toned down to focus on the contributions, presents the results comparing to well-tuned hyperparameters and not over-claim.

More comments:

What is the outer loop loss function? Is it assumed to be the same as the inner one (but using real data instead of training data)? I think this should be made explicit in the method section.

There are some additional experiments in other settings such as RL and unsupervised learning. Both seem like quite interesting directions but seem like preliminary experiments that don’t work convincingly yet. The RL experiment shows that using GTN does not change performance much. There is a claim about optimizing randomly initialized networks at each step, but the baseline which uses randomly initialized networks at each step with A2C is missing. The GAN experiments shows the GAN loss makes GTN realistic (as expected) but there are no quantitative results on mode collapse. (Another interesting experiment would be to show how adding a GAN loss for generating data affects the test performance of the method.) Perhaps it would benefit the paper to narrow in on supervised learning? Given that these final experiments are not polished, the claim in the abstract that the method is “a general approach that is applicable to supervised, unsupervised, and reinforcement learning” seems to be over-claiming. I understand it can be applicable but the paper has not really done the work to show this outside the supervised learning setting.

Minor comments:
Pg. 4: comperable -> comparable

**Experience Assessment:**

I have read many papers in this area.

**Review Assessment: Checking Correctness Of Derivations And Theory:**

I carefully checked the derivations and theory.

**Review Assessment: Checking Correctness Of Experiments:**

I carefully checked the experiments.

**Review Assessment: Thoroughness In Paper Reading:**

I read the paper thoroughly.

---

> ### Author Response · Authors · 2019-11-11
> **Response to review #2 (part 3)**
>
> > The GAN experiments shows the GAN loss makes GTN realistic (as expected) but there are no quantitative results on mode collapse.
>
> That is true, which is why we described them merely “encouraging initial results” and put them in the supplementary material instead of the main text. We do not think the GAN results are the main focus of the paper, but we could try to expand them (or cut them entirely) if you feel that is necessary and a better solution than what we do currently, which is simply letting the reader know about an intriguing possibility for future work.
>
> (Another interesting experiment would be to show how adding a GAN loss for generating data affects the test performance of the method.)
>
> Nice idea! We actually tried this at various times over the course of the research project. In general it hurts results. We think the reason is because GANs are incentivized to produce images that look like *real* images. Adding that realism constraint prevents the GTN from producing images that teach many concepts at once (e.g. about many different forms a 7 can take), but look unrealistic. Of course, that is just a hypothesis for why performance tended to be worse. We did not do systematic experiments, and GANs can be finicky (and there are many forms of them at this point) which is why we did not include the results in the paper. However, if you would like us to add them, we would be happy to. Please let us know.
>
> Perhaps it would benefit the paper to narrow in on supervised learning? Given that these final experiments are not polished, the claim in the abstract that the method is “a general approach that is applicable to supervised, unsupervised, and reinforcement learning” seems to be over-claiming. I understand it can be applicable but the paper has not really done the work to show this outside the supervised learning setting.
>
> We made the following change. Does this change address your concern? We are trying to balance between letting people know about the generality of the idea, but also making it clear that here we have only focused on the supervised case.
> Old: a general approach that is applicable to supervised, unsupervised, and reinforcement learning.
> New: a general approach that is, in theory, applicable to supervised, unsupervised, and reinforcement learning, although our experiments only focus on the supervised case.
>
> > Pg. 4: comperable -> comparable
>
> Fixed
>
> Thank you again for the constructive feedback! Please let us know if there are any other concerns you would like us to address. We hope you will consider increasing your score.

---

> > ### Comment · AnonReviewer2 · 2019-11-15
> > **Response to rebuttal**
> >
> > Thank you for your extensive response, it has helped clarify things a lot.
> >
> > 1. Regarding the #mainIssue, I understand now the motivation the paper but it is worth considering why there was a big misunderstanding. Perhaps some of the sections below can be ammended:
> >
> > Abstract: "GTNs are deep neural networks that generate data and/or training environments that a learner (e.g. a freshly initialized neural network) trains on before being tested on a target task. We then differentiate through the entire learning process via meta-gradients to update the GTN parameters to improve performance on the target task. GTNs have the beneficial property that they can theoretically generate any type of data or training environment, making their potential impact large. This paper introduces GTNs, discusses their potential, and showcases that they can substantially accelerate learning" - makes it sound like using GTNs can be applied to improve upon any nominal method on the target task, when in reality GTNs are for *quickly* learning on a target task, perhaps at the expense of performance. I understand "improves" here is talking about the optimization procedure, but this is easy to misinterpret.
> >
> > Abstract: "Overall, GTNs represent a first step toward the ambitious goal of algorithms that generate their own training data and, in doing so, open a variety of interesting new research questions and directions." - Seems to be speculative, and alludes to GTNs replacing all training datasets, when the paper really presents a tradeoff between using all the data vs a trained model that generates data.
> >
> > Page 2: first sentence in Methods. I see now you have changed this to include "rapidly" and this helps clarify.
> >
> > Page 4: " We next show that GTNs can generate a synthetic training set that is more effective than real training data in two supervised learning domains (MNIST and CIFAR10)" - it is not more effective in the usual sense (final performance of the model), I think you mean for a limited set of gradient steps, it is better.
> >
> > 2. The NAS results - I see now the significance of Table 1, and I think the performance on CIFAR is acceptably close to state of the art with much less CPU hours. I would very much like to see the same NAS results and correlations on MNIST as figure 3. I would also encourage the authors to release code so that people might try if the results hold on different datasets and with preprocessing tricks etc.
> >
> > 3. The curriculum description is now clear and the result is quite interesting that you can optimize these extra hyperparameters of the training process for a little extra performance.
> >
> > 4. The RL experiment - I understand this experiment now, and I think this is a very exciting experiment then and warrants further study (in future work). Perhaps it can be applied to good use in this setting: https://arxiv.org/abs/1812.02900
> >
> > In light of the rebuttal I am inclined to change my rating at the moment to "weak accept". As the rebuttal period is basically over, I would hope to see the comments above addressed in the camera ready if accepted.

---

> > > ### Author Response · Authors · 2019-11-15
> > > **Re: Response to Rebuttal**
> > >
> > > Dear Reviewer,
> > >
> > > Thank you for your response. We just made all of the specific changes you requested (see below for comment-by-comment responses) and have uploaded a revised PDF with those changes. We deeply appreciate your saying that you are inclined to change your score. That could very well make the difference between the paper being accepted and rejected. Either way, we sincerely appreciate the time and careful thought you put into your insightful reply and original review.
> > >
> > >
> > > > Comment: Thank you for your extensive response, it has helped clarify things a lot.
> > >
> > > > 1. Regarding the #mainIssue, I understand now the motivation the paper but it is worth considering why there was a big misunderstanding.
> > >
> > > We agree wholeheartedly. We believe the changes to the manuscript inspired by your comments in the original review and in your response are making the paper substantially more clear. Thank you for that.
> > >
> > > > Perhaps some of the sections below can be amended:
> > >
> > > Re: Abstract Issue #1
> > >
> > > We have changed the abstract to clarify this issue.
> > >
> > > Re: Abstract Issue #2
> > >
> > > We do mean in this sentence to speculate on first steps towards very ambitious future goals. To make that even more clear, we added “speculating forward” and “could”, resulting in the changed sentence being, “Speculating forward, GTNs may represent a first step toward the ambitious goal of algorithms that generate their own training data and, in doing so, open a variety of interesting new research questions and directions.” We hope that makes things clearer to the reader and addresses your concern, but please let us know if you prefer to see other or different changes.
> > >
> > > > Page 2: first sentence in Methods. I see now you have changed this to include "rapidly" and this helps clarify.
> > >
> > > Great
> > >
> > > > Page 4: " We next show that GTNs can generate a synthetic training set that is more effective than real training data in two supervised learning domains (MNIST and CIFAR10)" - it is not more effective in the usual sense (final performance of the model), I think you mean for a limited set of gradient steps, it is better.
> > >
> > > We made a change to make this much clearer. It now reads: “We next show that GTNs can generate a synthetic training set that enables more rapid learning in a few SGD steps than real training data in two supervised learning domains (MNIST and CIFAR10)”
> > >
> > > > 2. The NAS results - I see now the significance of Table 1, and I think the performance on CIFAR is acceptably close to state of the art with much less CPU hours. I would very much like to see the same NAS results and correlations on MNIST as figure 3. I would also encourage the authors to release code so that people might try if the results hold on different datasets and with preprocessing tricks etc.
> > >
> > > We will definitely release our code (and trained models, hyperparameters, etc.).
> > >
> > > As for MNIST, we are not aware of any major NAS papers that use MNIST as a benchmark, so we would not be able to compare our results to theirs. One reason no one seems to use MNIST could be because the problem is virtually solved, so there is no headroom for NAS to find even higher-performing architectures. Additionally, the task may be so easy that it would not showcase the difference in performance between different types of architectures, as so many of them can achieve high performance. We are happy to run MNIST NAS if you like, but we would not have any published results for other NAS algorithms to compare it with, and the results may be inconclusive for the reasons just mentioned.
> > >
> > >
> > > > 3. The curriculum description is now clear and the result is quite interesting that you can optimize these extra hyperparameters of the training process for a little extra performance.
> > >
> > > We are glad it is clear. We agree it is nice that we can get this performance lift without additional computation overhead.
> > >
> > > > 4. The RL experiment - I understand this experiment now, and I think this is a very exciting experiment then and warrants further study (in future work).
> > >
> > > We are delighted to hear that.
> > >
> > > > Perhaps it can be applied to good use in this setting <BCQ Paper Link>
> > >
> > > We love that paper. We agree that could dovetail nicely with GTNs.
> > >
> > > > In light of the rebuttal I am inclined to change my rating at the moment to "weak accept". As the rebuttal period is basically over, I would hope to see the comments above addressed in the camera ready if accepted.
> > >
> > > Thank you once again for your time and input. It has greatly strengthened the paper. All of these changes have been included in a just-uploaded revision to the PDF.

---

> ### Author Response · Authors · 2019-11-11
> **Response to review #2 (part 2)**
>
> > At a high level it would be very surprising to me if the way forward for better discriminative models was to learn good generative models and use them again for training discriminative models, simply because discriminative models have proved thus far significantly easier to train.
>
> We believe this comment is addressed by our response to the #mainIssue. Is that right? We are not claiming that training on synthetic data improves asymptotic performance over training on real data, just that it can speed-up training substantially (which is useful for NAS). As a side note: we do mention that an idea for future work is to see if GTNs can improve GANs by reducing mode collapse and provide some very preliminary experiments that are in line with that hypothesis, but we believe that is not what you are referring to here. Please let us know if we misunderstood your point here and we would be happy to address any other questions you have.
>
>
> > If this work does eventually show this result, it would be a very interesting result. At the moment, I believe it does not, but I would be happy to change my mind if the authors provide convincing evidence. Alternatively, I feel that the paper could be a valuable contribution to the community if the writing is toned down to focus on the contributions, presents the results comparing to well-tuned hyperparameters and not over-claim.
>
> We are glad you agree this paper makes a valuable contribution. We will try to change the writing in accordance with your suggestions. Specifically, we will make the issues in the #mainIssue clearer in the manuscript and also make it clear that our hypothesis that weight norm will help meta-learning in general is just that, a hypothesis that requires future work to validate. Would that (plus the changes we mention elsewhere) satisfy you, or are there other things you would like to see changed?
>
> > What is the outer loop loss function? Is it assumed to be the same as the inner one (but using real data instead of training data)? I think this should be made explicit in the method section.
>
> The outer loop loss function is domain specific (e.g. cross-entropy for logistic regression on real MNIST or CIFAR data). In our supervised learning experiments, the inner-loop losses match the outer-loop loss (but, as you note, with synthetic data instead of real data), but in the RL case, MSE is used in the inner-loop and an actor-critic loss is used in the outer-loop. We will update the text to make this clear.
>
> > There are some additional experiments in other settings such as RL and unsupervised learning. Both seem like quite interesting directions but seem like preliminary experiments that don’t work convincingly yet. The RL experiment shows that using GTN does not change performance much. There is a claim about optimizing randomly initialized networks at each step, but the baseline which uses randomly initialized networks at each step with A2C is missing.
>
> That plot for the RL results (Fig. 9) is actually very hard to read, in a way that is not good for showcasing the benefits of GTN. At every point in that plot, we are comparing the performance of a new, randomly initialized architecture trained by synthetic GTN data **from scratch in one step of SGD** vs. the performance of the A2C network trained *cumulatively* up to that point. Thus, at the 100,000th step of training, we can train a new architecture to the given performance (of around 190) 100,000 times faster with GTN synthetic data than A2C from scratch! We will make this clearer in the text. Of course, this is a very simple RL problem, but it is still thought-provoking, which is why we think readers will benefit from having it in the paper. Do you agree?
>
> We are not sure what you mean by the randomly initialized networks baseline: starting A2C from a randomly initialized network is what is shown in the red A2C Agent curve. If you reset that at each step, the performance would stay flat at near zero because A2C’s performance after one SGD step is effectively zero.

---

> ### Author Response · Authors · 2019-11-11
> **Response to review #2 (part 1)**
>
> Thank you for the valuable and thoughtful feedback. We are glad you think our approach is interesting. We will address and clarify all of your concerns in our revision.
>
> > To me, the main claim is very surprising and counter-intuitive - it is not clear where the extra juice is coming from, as the algorithm does not assume any extra information. The actual results I believe do not bear out this claim because the actual results on MNIST and CIFAR10 are significantly below state of the art.
>
> Please see our #mainIssue response to all reviewers.
>
> > This does not negate the potential of GTNs which I feel are an interesting approach, but I believe the paper should be more straightforward with the presentation of these results.  The current results basically  show that GTNs improve the performance of learners with bad hyper-parameters. On problems that are not as well-studied as MNIST or CIFAR10 this could still be very valuable (as we do not know what performance is good or bad in advance).
>
> We thank the reviewer for recognizing this advantage. We note that we do optimize the hyperparameters for the controls (and GTN), but we agree that we are not using SOTA architectures during GTN training. In light of our response on the main issue above (see #mainIssue), we hope it makes a bit more sense why we are doing that and why our results are important despite that. They show that, for a given distribution of architectures, training on synthetic data is much *faster* than real data (not that it leads to better accuracy). It does not matter (for NAS, the purpose of the paper) whether the underlying distribution of architectures are great architectures (that produce SOTA results) or mediocre architectures. In fact, for NAS we need them to be diverse (and thus not just represent the highest-performing architectures) because we want the GTN-estimates to generalize to many types of architectures. That said, it would be a problem if GTN were unable to predict that great architectures are better than mediocre architectures (because that would hurt the ability of GTN-NAS to discover those architectures during search), but our results in the NAS section show that GTN predictions do generalize, and do so especially well for high-performing architectures. See Fig. 4c, which shows a high correlation between GTN estimates and the ground truth for high-performing architectures. Moreover, GTN-NAS does in fact find high-performing architectures (Table 1), so clearly GTN’s estimates are accurate for some high-performing architectures. Note that GTN does not have to predict that every high-performing architecture is high performing; rather, to be successful it only needs to find a subset of all possible high-performing architectures, and the results confirm that it is able to do so.
>
> > Based on the results, GTN does seem to be a significant step forward in synthetic data generation for learning compared to prior work (Zhang 2018, Luo 2018).
>
> Thanks for bringing that up. We think we made significant progress in the synthetic data for learning direction (Maclaurin et al., 2015, Wang et al., 2019b) as well as learning to teach overall (Fan et al., 2018). We believe we are the first to apply these methods to NAS and obtain results competitive to other efficient NAS methods (Zhang 2018, Luo 2018).
>
> > The paper proposes two other contributions: using weight normalization for meta-learning and curriculum learning for GTNs. Weight normalization is shown to stabilize GTNs on MNIST. I think the paper oversteps in the relevant method section, hypothesizing it may stabilize meta-learning more broadly. The paper should present a wider set of experiments to make this claim convincing. But the point for GTNs on MNIST nevertheless stands.
>
> When we first mention this hypothesis, we will add text that makes it clear that it is a hypothesis only, and that future work with other meta-learning algorithms is needed to provide evidence for this claim. We already say exactly that in the discussion (the only other time it is mentioned). Would that suffice? We do want to alert readers to this possibility in methods (in case they do not make it to the discussion) because it is a simple technique to try and it could help them with their meta-learning research.
>
> > For curriculum learning: the description of the method is done across section 2 and section 3.2 and does not really describe it completely. How exactly are the samples chosen in GTN - All Shuffled? How does GTN - Full Curriculum and Shuffled Batch parametrize the order of the samples so that it can be learned? I suggest that this information is all included as a subsection in the method (section 2). The results seem to show the learned curriculum is superior to no curriculum.
>
> We will update the paper to contain a better explanation of the curricula.  We will reply here once that updated, clearer text is ready.

---

### Author Response · Authors · 2019-11-11
**Response to main issue**

We would like to thank all reviewers for their insightful comments! We recognize that reviewing is time-consuming work, and we are deeply appreciative. Below, we’ve written responses to each reviewer individually. We also plan to upload a revised version of the paper in the next few days that will implement the suggested changes. Thank you once again for your time and assistance!

#mainIssue
All three reviewers raised a similar, central issue. We thus want to address that issue here in the reply to all. We give it the tag #mainIssue because we also refer to our response to this issue at times in our individual responses to each reviewer.

All three reviewers commented that our MNIST and CIFAR performance does not match the state-of-the-art (SOTA) on those benchmarks. We apologize for the miscommunication, for we failed to communicate that SOTA performance on those benchmarks when training with GTN-generated data is not an important metric for our technique. We are currently improving the writing to clarify this issue and will upload a new draft in a few days. Here is an explanation of why that is not the right metric to focus on for GTNs.

Because the ultimate use of GTNs is to accelerate neural architecture search (NAS), what matters is *not* the accuracy that can be achieved at convergence with many steps of SGD on synthetic data, but instead how well we can identify architectures that have high asymptotic performance (when trained on real data). A means to that end is being able to train architectures rapidly, i.e. with very few SGD steps, because doing so allows us to rapidly identify promising architectures for NAS. To clarify things, we will call training until convergence asymptotic performance, and we will describe the idea of k-step-performance, where k is low (e.g. 32 or 128 steps). We will call this “few-step performance.” With this terminology, we can revisit our overall methodology: We train a GTN to produce synthetic data that produces high few-step performance on a distribution of “training” architectures. We then launch NAS, and for each newly generated architecture we perform a few steps of SGD with GTN-produced synthetic data, which rapidly provides an estimate of that architecture’s true, asymptotic performance (when trained on real data). That estimate fuels NAS. We then take the best architecture found according to this GTN-produced estimate, and train that to convergence on real data. The true tests of the method are (1) how high the asymptotic performance of the best architecture found is when trained on real data and (2) how little computation is required to discover that architecture. Both are reported in Table 1, and our method is competitive with SOTA NAS methods, but via an entirely new technique, offering our community an interesting, useful new tool.

With that recap, we agree with you that training a learner with GTN-produced synthetic data does not produce SOTA asymptotic performance for that learner, but that is not relevant for the NAS task. The important discovery we show is that GTNs are SOTA for few-step performance (i.e. <= 128 SGD steps). That discovery enables rapidly training a newly sampled architecture to estimate its true (asymptotic) performance, which accelerates NAS. To reiterate,  the main point of the paper is to produce an entirely new NAS method that is competitive with the NAS-SOTA, not to produce actual SOTA results on MNIST and CIFAR.

One additional note is that there are at least two major reasons for the gap between the performance of learners trained on GTN-produced synthetic data on MNIST and CIFAR and the SOTA on those benchmarks: (1) SOTA results are with asymptotic performance (i.e. many, many steps of SGD), as just discussed, and (2) SOTA results are with better architectures specifically selected after much human effort to produce SOTA results on those benchmarks. Our method averages over a distribution of architectures that are purposefully chosen to be diverse (and thus not a small set of SOTA architectures) because for NAS-purposes we want the GTN few-step performance estimates to generalize to new architectures (in order to find better architectures).

---

> ### Author Response · Authors · 2019-11-11
> **Response to main issue (part 2)**
>
> Now, you could rightfully ask whether the best architecture discovered by our NAS method (GTN-NAS) achieves asymptotic SOTA performance (when trained on real data, which is what one ultimately should do with GTN-NAS-discovered architectures) . As reported in Table 1, the architecture GTN-NAS finds is competitive with architectures discovered by other SOTA-NAS methods (Luo et al. 2018, Zhang et al. 2018). Additionally, the asymptotic performance of the GTN-NAS-discovered architecture (when trained on real data), is close to the actual CIFAR10 SOTA (2.42 for GTN vs. 2.08% for the SOTA, according to https://paperswithcode.com/sota/image-classification-on-cifar-10). However, the actual SOTA on these benchmarks relies on tricks we did not take advantage of, such as training on data outside of these datasets, complex data augmentation techniques, etc. We think it is sufficient to show that one can find a very good architecture with GTNs in a way competitive with the SOTA, but via a very different technique (adding a new tool to our community’s toolbox). We do not think it is necessary that the discovered architecture itself is the SOTA on these benchmarks. As some reviewers noted, this capability to automatically discover near-optimal architectures will be useful in new domains wherein we as a community have not yet figured out (at great cost) what the top-performing architectures are.
>
> We will modify the paper to make all of this much more clear.
>
> We also want to address the question of where the “extra juice” (as R2 put it) is coming from. We agree that one would not expect that training a generator to produce synthetic data to encourage a learner to do well on real data would lead to better performance than training directly on the real data. While such a counter-intuitive scenario is not inconceivable (if, for example, the generator provides some useful regularization for some reason, such as it having few parameters and good priors baked into it), that is *not* what we are claiming is happening, should happen, or that we have evidence for. Instead, we are saying (and showing) that it is possible to *compress* the information contained in a large set of real world samples into a small set of synthetic samples (the ones produced by the GTN). Thus, no information is created, but it is instead compressed. In some sense, this achievement does not require anything remarkable: we can imagine the original dataset might contain many similar images, where only a few of them would be sufficient for training. Learning to select such a useful subset of the dataset has been shown to speed up training, and we cite papers in our paper that have shown that. Here the GTN is *learning* to do that compression, but is doing it with the extra degrees of freedom of being able to generate arbitrary synthetic images (so it could combine many different things that need to be learned about images into one image, for example).
>
> Given all of these explanations of the different motivations of our paper and why SOTA on CIFAR and MNIST is not a relevant metric (and that we are not proposing to produce better asymptotic performance than the real data), would you be willing to increase your score for our paper? It does produce an entirely new, interesting type of NAS method that is competitive with the SOTA for NAS. Your consideration of whether to increase the score is deeply appreciated.

---

### Author Response · Authors · 2019-11-14
**Updated PDF**

Dear reviewers. We promised we would post an updated PDF with the changes we said we would make. We have just uploaded that PDF. Please let us know if you would like to see any further or different changes. The only set of changes we have not yet made regard the issue of the images being alien and unrecognizable. We are waiting to hear whether you think we should add our hypotheses for why that might be the case to an SI section. Additionally, we are actively trying some experiments that might shed additional light on this issue, and those experiments are ongoing. We will let you know if anything conclusive emerges from them.

Thanks, as always, for your time. We know it is extremely valuable.
GTN Authors

---

### Decision · Program_Chairs · 2019-12-19

**Decision:**

Reject

**Comment:**

Overview:
This paper introduces a method to distill a large dataset into a smaller one that allows for faster training. The main application of this technique being studied is neural architecture search, which can be sped up by quickly evaluating architectures on the generated data rather than slowly evaluating them on the original data.

Summary of discussion:
During the discussion period, the authors appear to have updated the paper quite a bit, leading to the reviewers feeling more positive about it now than in the beginning. In particular, in the beginning, it appears to have been unclear that the distillation is merely used as a speedup trick, not to generate additional information out of thin air. The reviewers' scores left the paper below the decision boundary, but closely enough so that I read it myself.

My own judgement:
I like the idea, which I find very novel. However, I have to push back on the authors' claims about their good performance in NAS. This has several reasons:

1. In contrast to what is claimed by the authors, the comparison to graph hypernetworks (Zhang et al) is not fair, since the authors used a different protocol: Zhang et al sampled 800 networks and reported the performance (mean +/- std) of the 10 judged to be best by the hypernetwork. In contrast, the authors of the current paper sampled 1000 networks and reported the performance of the single one judged to be best. They repeated this procedure 5 times to get mean +/- std. The best architecture of 1000 is of course more likely to be strong than the average of the top 10 of 800.

2. The comparison to random search with weight sharing (here: 3.92% error) does not appear fair. The cited paper in Table 1 is *not* the paper introducing random search + weight sharing, but the neural architecture optimization paper. The original one reported an error of 2.85% +/- 0.08% with 4.3M params. That paper also has the full source code available, so the authors could have performed a true apples-to-apples comparison.

3. The authors' method requires an additional (one-time) cost for actually creating the 'fake' training data, so their runtimes should be increased by the 8h required for that.

4. The fact that the authors achieve 2.42% error doesn't mean much; that result is just based on scaling the network up to 100M params. (The network obtained by random search also achieves 2.51%.)

As it stands, I cannot judge whether the authors' approach yields strong performance for NAS. In order to allow that conclusion, the authors would have to compare to another method based on the same underlying code base and experimental protocol. Also, the authors do not make code available at this time. Their method has a lot of bells and whistles, and I do not expect that I could reproduce it. They promise code, but it is unclear what this would include: the generated training data, code for training the networks, code for the meta-approach, etc? This would have been much easier to judge had the authors made the code available in anonymized fashion during the review.

Because of these reasons, in terms of making progress on NAS, the paper does not quite clear the bar for me. The authors also evaluated their method in several other scenarios, including reinforcement learning. These results appear to be very promising, but largely preliminary due to lack of time in the rebuttal phase.

Recommendation:
The paper is very novel and the results appear very promising, but they are also somewhat preliminary. The reviewers' scores leave the paper just below the acceptance threshold and my own borderline judgement is not positive enough to overrule this. I believe that some more time, and one more iteration of reorganization and review, would allow this paper to ripen into a very strong paper. For a resubmission to the next venue, I would recommend to either perform an apples-to-apples comparison for NAS or reorganize and just use NAS as one of several equally-weighted possible applications. In the current form, I believe the paper is not using its full potential.